# Explanation of the Influence of Sodium Chloride Solution on Volume Deformation and Permeability of Normally Consolidated Clays

**DOI:** 10.3390/ma12101671

**Published:** 2019-05-22

**Authors:** Tongwei Zhang, Shijun Wang

**Affiliations:** 1School of Civil Engineering and Mechanics, Lanzhou University, Tianshui Road 222, Lan Zhou 730000, China; 2Key Laboratory of Mechanics on Disaster and Environment in Western China, Lanzhou University, Tianshui Road 222, Lan Zhou 730000, China; 3Economy and Technology Research Institute, Gansu Electric Power Company, State Grid, Lanzhou 730050, China; angelofkill@126.com

**Keywords:** sodium chloride solution, bentonite, kaolin, volume deformation behavior, permeability, microstructure

## Abstract

The marine clays located in the Southeast area of China are characterized by their higher water content, higher compressibility and higher salinities. This soil is mainly composed of illite/montmorillonite interlayer minerals. Previous research has shown that the saline water significantly influences the liquid limit and other physical characteristics of the clays. As the desalination of pore water occurs as a result of freshwater or rainfalls, the physical and mechanical behaviors of the soft marine clays changes, and this can lead to potential hazards for infrastructure. Therefore, it is essential to understand the effects of chemistry variations and to predict the long-term foundation deformations. Based on previous works, the deformation behavior of artificial soils corresponding to a mixture of kaolinite and bentonite (the mass ratios of bentonite were 0%, 5%, 10% and 20%) was further discussed in a ln(1 + *e*)-log*p’* system. The permeabilities of the samples mixed with different concentrations of sodium chloride solutions were compared based on oedometer tests. The micro-structures in the samples were investigated by SEM (scanning electronic microscopy) tests. The declining trend of a newly defined volume compression index *C*_cv_ and swelling index *C*_sv_ with pore water salinity and *e*_0_/*e*_L_ was observed when the initial void ratios *e*_0_ of the samples were close. The permeability coefficient *k* and the slopes *C*_k_ = *e*/log*k* of the mixtures increased with the ionic concentrations. Finally, the changes in volume deformations and permeabilities induced by sodium chloride solution are discussed based on ‘suction pressure’ and initial void compression at micro-level. This paper proves that the influences of salinity on the mechanical behavior of clays are mainly attributable to the interaction between diffused double layers, and these findings are helpful for improving the constitutive model of soft clays when taking pore water chemistry changes into consideration.

## 1. Introduction

The hydro-chemo-mechanical coupling effect of very active soils has drawn increasing attention with regard to the geotechnical problem of soil-environment interactions and the synthesis of composite materials. For example, the pore water chemical effect on compacted bentonite is of significant importance in the buttering of waste landfills and repositories for high levels of radioactive waste (HLW). Recently, organ-modified montmorillonite was used as filler for nanocomposite resins [1]. Meanwhile, salinity variations in pore water can also change the engineering properties of marine clays [2]. Marine soft clays are quaternary sediments and are widely distributed in southeastern China. Since the economic development of this region and the springing up of substantial infrastructure projects, the mechanical behavior of marine clay in relation to the environment has been drawing increasing attention. Marine clays are deposited in sea regression environments, and high water salinity and montmorillonite content are their main characteristics [2].

One of the earliest papers studying the sensitivity of physical properties such as the liquid and plastic limits of clay soil to ionic types in pore water was published by Winterkorn and Moorman (1941) [3]. Bolt (1956) proposed that the compressibility of bentonite suspensions can be well described by considering the interaction between the electric double layers around particles [4]. The mineral composition effect on residual shear strength was ascertained by drained direct-shear tests in the 1960s [5]. The study on settlements of buildings in Drammen using vane shear test showed that long-term leaching by fresh water had a dramatic effect on the properties of the marine clay [6]. Based on diffuse double layer theory, Mitchell (1976) proposed that the increase of pore fluid’s ion concentration would cause a reduction in the half-space distance in double diffusion layer systems [7]. The noticeable impact of pore liquid chemistry on the compression behavior of very active smectitic clay such as Bisaccia clay and Marino clay was proposed in 1990s [8,9]. Kaya et al. (2006) focused on the between-sedimentation volume of clays and the dielectric constant of fluid [10]. Di Maio et al. (2004) reported that the intrusion of solutions caused a reduction in the deformation of compacted bentonite [11]. Gajo and Maines (2007) investigated the mechanical behavior of active clays affected by pore fluid acidity and alkalinity [12]. Recently, many researchers have focused on the influence of infiltrating solutions coupled with rock fracture on the water retention curves, swelling behavior, hydraulic conductivity, and yield stresses of highly compacted bentonite [13,14,15,16,17]. Most previous studies have been conducted with reference to pure unsaturated bentonite or active clays prepared by different procedures, but the chemical effect on saturated remolded clay has rarely been mentioned.

In this paper, we focus on the clay mineral composition and salinity effect on volume compression-swelling behavior, permeabilities and microstructures of saturated remolded soil with a water content more than the liquid limits. The varieties of mineral composition and pore water salinity used in this study were based on a geological survey of Lianyungang marine clay in coastal areas. With the help of ‘suction pressure’ and micro-structure observations, a reasonable mechanism of the test results was suggested.

## 2. Experimental Method

### 2.1. Material and Sample Preparation

The two commercial clays (kaolinite and bentonite) and one natural clay taken from the Southeast of China were used in this paper. The basic physical properties, such as the specific gravity, limit moisture content, specific surface area (SSA) and clay minerals of these soils, were introduced by Deng et al. (2014, 2018) [2,18]. Laser-based techniques for particle-size measurement were used and the results are reported by Figure 1. The procedure of laser analysis is summarized as follows: (1) 30 g of air-dried soil was stirred with 200 mL distilled water or sodium chloride solution (a mass concentration of 5%) and then soaked for 16 h; (2) the suspension was then slowly added dropwise into the container of the laser analyzer (MAF5000, Malvern Instruments Ltd., Malvern, UK), which was filled with 800 mL distilled water; (3) this test was terminated when the required attenuation (10~20%) of the laser beam was reached; (4) based on the relationship between the scattering angle, particle radius and angular distribution of light intensity, the particle size distributions of the fine particles were obtained. Due to the limitations of light visibility, particles smaller than 0.3 μm could not be detected.

Based on the properties of the in situ clay sampled from Lianyungang, the distilled water and sodium chloride solutions at mass contents of 1%, 3%, 5% and 10% were mixed with kaolin and kaolin-bentonite mixtures, and the ratio of bentonite was 5%, 10% and 20%. Hereafter, ‘B’ and ‘K’ will be used to refer to ‘bentonite’ and ‘kaolin’, respectively, and the following percentage will represent their contents, and the data after the plus sign will be the sodium chloride concentration. The soils in their dry state and the solutions were thoroughly stirred and sealed for 24 h to homogenize the distribution of the pore liquid. Then, the slurries were filled into the oedometer ring. The initial water contents and void ratios are listed in Table 1. After that, the specimens were installed in the special oedometer cell, then soaked in the sodium chloride solutions as shown in Figure 2. To avoid the solution being contaminated by the metal container, the apparatus was made of organic plastic

### 2.2. Apparatus and Test Method

The vertical loads 1 kPa, 1.8 kPa, 3.2 kPa and 5.8 kPa were applied first, to avoid squeezing the soil from the oedometer. Then, a load of 12.5 kPa was applied, and the following doubled loads were applied step by step until 1600 kPa was reached, following the standard procedure in BS1377-2 (1990). Each step of loading was held constant for at least 24 h. Once 1600 kPa had been loaded, each addition was removed stepwise until 12.5 kPa had been reached, following the same sequence. Then, the volume compression-swelling curves ln(1 + *e*) − log*p* and calculated permeability based on consolidation theory were evaluated.

The samples ‘B20%K80%Micro’ and ‘KMicro’ were used in SEM (scanning electronic microscopy) tests, aiming to clarify the mechanism of salinity effects on the deformation behavior of soft clay. As illustrated in Table 1, the mixtures were remolded with distilled water (B20%K80%Micro + 0% and KMicro + 0%) and 5% NaCl solution (B20%K80%Micro-02 and KMicro-02) at 1.2*LL* (*w*_0_ = 63%) and 1.3*LL* (*w*_0_ = 43%), similar to the samples used in the oedometer tests. Samples were consolidated until 100 kPa, and then carefully trimmed to cubes with a size of 1 cm × 1 cm × 1 cm. Then, the blocks were immersed in liquid nitrogen and coated with a layer of gold to conduct SEM tests [19,20,21].

## 3. Results and Discussion

### 3.1. Physical Properties

The thickness of montmorillonite layers is about 0.96 nm, and the thickness of kaolinite is about 50 nm to 800 nm [22]. As per previous studies, clay lamellae often combine together into particles, usually in the form of stacks, tactoids or quasicrystals [23]. The clay mineral compositions of B, K and LYG clays obtained from XRD (X-ray Diffraction) analysis were 95%, 68% and 45% [1,17]. The clay fractions (particles smaller in size than 0.002 mm, BS1377-1990) of B, K and LYG clays based on laser-based particle-size measurements were about 20%, and the proportion of fine fractions (particles passing through a 0.063 mm test sieve) was 90% for B and K clays, and 70% for LYG clay. This indicated that the clay lamellae may flocculate and exist in the form of stacks, tactoids or quasicrystals.

The liquid limits and plasticity index were evaluated by mixing the soils with distilled water and NaCl solutions, and the results were plotted in Figure 3. The liquid limits of commercial kaolinite containing different salinities were close. Based on the ASTM standard D2487 (2010), three tested materials (K, B5%K95% and B10%K90%) were low-plasticity clays (CL), whereas LYG clay and B20%K80% were high-plasticity clays (CH). The LYG clay and B20%K80% reconstituted with salt solutions changed from high-plasticity clays to low-plasticity clays. This can be explained by the fact that the liquid limit of montmorillonitic soils (such as bentonite) is mainly controlled by the diffuse double layer of water surrounding the clay particles [11,24], while that of kaolinitic soils is primarily determined by the interparticle forces, such as the fabric [25,26].

### 3.2. Volume Compression and Swelling Index

It is widely known that the volume of soil is represented by (1 + *e*), where *e* is the void ratio of soil. Butterfield (1981) first proposed that the consolidation test results can be well interpreted by two linear lines in the ln(1 + *e*)-log*p’* plot, where *p’* is the effective applied stresses [27]. Onitsuka et al. (1995) verified the consistency of the ln(1 + *e*)-log*p’* method for undisturbed Ariake clays [28]. The stress at the intersection point of bilinear lines in the ln(1 + *e*)-log*p’* is the consolidation yield stress, and the bilinear lines are respectively termed the pre-yield and post-yield lines. The *e*-log*p’* curve of a completely remolded soil (*w*_0_ = 1.0~1.5 *w_L_*) is shaped slightly convex downwards [29], but they can be well defined by a straight line after the yielding stress in the bilogarithnic curve of ln(1 + *e*) versus log*p’* [30]. The *e*-log*p’* curves of the remolded bentonite-kaolin mixtures were reported by Deng et al. (2018) [17], and their compression-swelling behavior was further discussed in ln(1 + *e*)-log*p’* systems.

Firstly, the volume compression index *C*_cv_ and swelling index *C*_sv_ are defined as follows:
(1)$$C_{cv}(C_{sv})=\Delta \ln(1+e)/\Delta \log p^{\prime }$$

Burland (1990) proposed the intrinsic compression index *C*^*^*_c_* and void ratios *e*^*^_100_, *e*^*^_1000_ corresponding to *p =* 100 kPa, 1000 kPa respectively. In this paper, the volume compression index was represented by the slope of straight part in ln(1 + *e*)-log*p’* curves between 100 and 1000 kPa, which is *C*_cv_ = ln(1 + *e*_100_) − ln(1 + *e*_1000_). The volume swelling index was defined by the slope of the line connecting the starting and ending point of swelling curves such as *C*_sv_ = (ln(1 + *e*_1600_) − ln(1 + *e*_12.5_))/log(1600/12.5).

The volume compression index and swelling index changing with different pore water salinities were presented in Figure 4. The *C*_cv_ and *C*_sv_ of B5%K95%, B10%K90% and B20%K80% significantly decreased with the increase in salt solution, and the decrements of *C*_cv_ and *C*_sv_ depending on salinities were larger when the bentonite content increased. Meanwhile, the *C*_cv_ and *C*_sv_ of K were within the ranges of 0.15 to 0.17, and 0.02 to 0.03, respectively. The salinity effect on the volume change of kaolin was not noticeable.

Cerato and Lutenegger (2004) illustrated that the initial water content *w*_0_ can influence the intrinsic properties *C^*^_c_* and *e^*^*_100_ [31]. Also, the pre-stresses significantly influenced the mechanical behavior of composite materials [32]. For soils, Hong et al. (2007) proposed that the proportion of initial void ratio *e*_0_ to the void ratio at the liquid limit *e*_L_ can be used for estimating a ‘suction pressure’ (*σ*_s_’) if the oedometer test data before yield stress was absent [33]. The ‘suction pressure’ (*σ*_s_’) of remolded clay is assumed to be the stress at the intersection point of bilinear lines in the ln(1 + *e*)-log*p’* plots. Based on the oedometer tests of samples whose initial water contents were 0.7~2.0 times the liquid limits, Hong et al. (2010) proposed an equation to calculate suction pressure (*σ*_s_’) as follows:(2)$$\sigma_{s}^{\prime }(kPa)=5.66/{\left(e_{0}/e_{L}\right)}^{2}$$

As mentioned, the liquid limits *w*_L_ of clays rich in smectite decreased with the NaCl concentrations, while those of the kaolinite were relatively stable. Then, based on Equation (2), the ‘suction pressure’ (*σ*_s_’) should theoretically decrease with *e*_0_*/e*_L_ = *w*_0_*/w*_L_ and NaCl concentrations in clays with an identified initial water content. The curves of *e*_0_*/e*_L_ versus *C*_cv_ or *C*_sv_ are presented in Figure 5, and a declining trend was observed when the initial void ratios *e*_0_ of the samples were close.

### 3.3. Permeabilities

The permeability coefficient is an important parameter reflecting the mechanical properties of soil. It is closely related to engineering problems such as foundation deformation and groundwater seepage. Many scholars have shown that the variation in the permeability coefficient *k* during the consolidation process is related to the void ratio *e*. Taylor (1948) proposed that the permeability coefficient *k* had a linear relationship with the change in the void ratio *e*.
(3)$$\mathrm{lg}k=\mathrm{lg}k_{0}-\frac{e_{0}-e}{C_{k}}$$
where *e*_0_ is the void ratio of soil at the initial state; *k*_0_ is the permeability coefficient at the void ratio *e*_0_, and *C*_k_ is a constant representing the ratio of logarithmic increment of the permeability coefficient to the increment of the void ratio. Mesri and Olson (1971) investigated the permeabilities of soils with different void ratios, and the experimental results showed that there was a linear relationship between lg*k* and *e* for one soil.

Based on Taylor’s theory, the parameter *C*_k_ was defined as the slope of *e*-lg*k* curve when the strain was less than 20%. After laboratory tests on soils with initial void ratios ranging from 0.8 to 2.0, Tavenas et al. (1983) suggested that the parameter *C*_k_ is proportional to the initial void ratio *e*_0_, and the relationships between void ratio *e* and permeability coefficient *k* changed with clay types. However, few studies have considered the effect of pore water chemistry on the permeability coefficient of clays.

The salinity effects on the elapsed period of primary consolidation and consolidation coefficient *C*_v_ of pure clays were investigated by Zhang et al. (2018. 2019) [18,30]. The results showed that the primary consolidation process of bentonite-kaolin mixtures became shorter under the impact of water salinity. To calculate the permeability coefficient, the time *t*_50_ when the consolidation degree reached 50% was firstly determined using the Cassagrand method based on the *e*-lg*t* curves. Then, the consolidation coefficient *C*_v_ can be calculated using the following formula:
(4)$$C_{v}=\frac{0.197H^{2}}{t_{50}}$$
where *H* is the maximum drainage distance, which is the average height of the specimen under loading. According to the Terzaghi’s one-dimensional consolidation theory, the permeability coefficient *k* can be calculated by the following formula:
(5)$$k=\frac{C_{v}\gamma_{w}a}{1+e}=\frac{C_{v}\gamma_{w}}{E_{s}}$$
where *a* is the compression coefficient and *E*_s_ is the compression modulus.

The relationships between log*k* and void ratio *e* for samples with different salinities are shown in Figure 6. The results show that the logarithm of permeability coefficient increased linearly with the increase of void ratio. The changes in the slope *C*_k_ = *e*/log*k* of the fitting line defined by Tavenas et al. (1983) with NaCl concentration are presented in Figure 7. For kaolinite, the permeability coefficient *k* of samples with an identified initial void ratio showed negligible differences when the pore water salinities were changed. In the meantime, the slopes of the fitting line represented by the *C*_k_ were close. When the bentonite fraction increased, the permeability coefficient significantly changed with pore water chemistry. For B5% K95% with the similar initial void ratio, the larger the pore water salinities were, the larger the permeability coefficient was. The slopes of the fitting line *C*_k_ decreased with the increase of saline concentration. The samples B10% K90% and B20% K80% followed a similar law as B5% K95%, but the differences between permeability coefficients *k* and *C*_k_ were larger than that attributed to the increase in salinity.

Therefore, the widely accepted knowledge that the coefficient *C*_k_ is linearly related to the initial void ratio was extended. The prediction for the permeability coefficient *k* in pure clays containing montmorillonite should consider the influence of pore water salinity.

### 3.4. Microstructures and Mechanism

The effect of salinity on microstructure was examined by SEM images (magnified 500 times and 5000 times), as shown in Figure 8 and Figure 9. Pores larger than 100 μm in the samples were not identified in Figure 8a–d, and the differences between their appearances were not obvious at a scale of 500 times magnification. When the kaolinite was magnified to 5000 times in Figure 9a,b, KMicro + 0% and KMicro + 5% showed a similar particle arrangement, and their fabric elements presented a bulky pellet shape. The diameter of their aggregates ranged from 10 to 20 μm, and the diameter of the pores ranged from 5 to 10 μm. The influence of water chemistry on the diameters of the aggregates and pores was not significant for specimens composed only of kaolin.

On the other hand, the fabric element of specimen B20%K80%Micro + 0% hydrated with distilled water (Figure 9c) exhibits a thin platelet shape, while that of specimen B20%K80%Micro + 5% salinized with 5% NaCl solution (Figure 9d) presents a bulky pellet shape. The lengths of the flakes of sample B20%K80% + 0% ranged from 10 μm to 40 μm, and the thickness was less than 1 μm. The diameters of the bulky pellets of sample B20%K80%Micro + 5% were also located in the range of 10 μm~40 μm. The pore sizes of B20%K80%Micro + 0% and B20%K80%Micro + 5% were close. Therefore, the pore water salinity caused the palte-like structures of the rich montmorillonite clays to change to blocky structures. Therefore, a flocculation mechanism could be introduced to interpret the effect of salinity on volume deformation for reconstituted clay, especially in those with rich montmorillonite.

As mentioned, the increase of the pore fluid’s ion concentration will cause a reduction in the half-space distance in double diffusion layer systems, according to diffused double layer theory. In the meantime, when the double diffusion layer is compressed, the decrease in repulsive forces is greater than that in attractive forces. Consequently, the fine-grained soils prefer to be flocculated [34]. Sridharan and Prakash (1999) considered the double diffusion layer theory to be more applicable for soils rich in montmorillonite than soils rich in kaolinite. Therefore, the salinity effect on the flocculation of clayey particles in B20%K80%Micro can be observed in Figure 9. On the other hand, Mesri and Olson (1971) considered that there were two factors controlling the permeability: (1) the sizes and shapes of particles; (2) the arrangement of particles. For the soils at flocculation state, the relatively large flow channels controlled the permeabilities. For soils in the dispersion state, the nearly homogenized pores caused curved flow channels. Then, the larger pores of samples B5%K95%, B10%K90% and B20%K80% in saline environment increased their permeabilities.

Figure 10 is a conceptual diagram to explain the results in this study. In the beginning, the effective void compression induced by the salt solution between the interlayer occurred before the consolidation process. Soils with higher salinity are more flocculated and aggregated than those with lower or without salinity, especially in soils rich in montmorillonite. The suction pressure decreased with increasing *e*_0_/*e*_L_ based on Equation (3). Both the pre-yield state and swelling state can be considered elastic deformation, so the compression curve of the pre-yield state in ln(1 + *e*)-log*p* coordinates was assumed to have the same slope during the rebound process. Then, the slope before and after suction pressure changed as shown in Figure 5.

## 4. Conclusions

To better understand the salinity effect on the volume compression and swelling behavior of clays, oedometer tests and SEM (Scanning electronic microscopy) tests were conducted on four artificial clays. Several conclusions can be drawn from this paper:
The volume compression and swelling behavior were discussed in ln(1 + e)-log*p’* coordinates. The re-defined volume compression index *C*_cv_ and swelling index *C*_sv_ of montmorillonite-rich clays significantly decreased with increasing salt solution, and the decrements of *C*_cv_ and *C*_sv_ depending on salinities were larger when the bentonite content increased. Meanwhile, the *C*_cv_ and *C*_sv_ of kaolin were within a narrow range when the salinity changed. For the bentonite-kaolin mixtures, a declining trend of *C*_cv_ and *C*_sv_ with *e*_0_/*e*_L_ was observed when their initial void ratios were close.For kaolinite, the permeability coefficients *k* and the slopes *C*_k_ = *e*/log*k* of samples had negligible differences when the pore water salinities changed. When the bentonite fraction increased, the permeability coefficients and *C*_k_ significantly decreased with pore water salinity.The SEM observation shows that the fabric element of bentonite-kaolin mixtures with distilled water forms a thin platelet shape, while that with 5% NaCl forms a bulky pellet shape. On the other hand, for the soils composed only of kaolinic mineral, the arrangement and pattern of the clay particles remain relative stable with pore water salinity.The interlayer compression induced by the salt solution, which has been proved by previous studies, occurs before the consolidation process. Meanwhile, the ‘suction pressure’ theoretically decreases with increasing *e*_0_*/e*_L_. Then, the compression and swelling curves in the ln(1 + *e*)-log*p* system changed with water salinity.

This paper proved that the influences of ionic concentration on the mechanical behavior of clays are mainly attributable to particle flocculation, and the conclusions are helpful in improving the constitutive model of soft clays in consideration of pore water chemistry changes.

## Figures and Tables

**Figure 1 materials-12-01671-f001:**
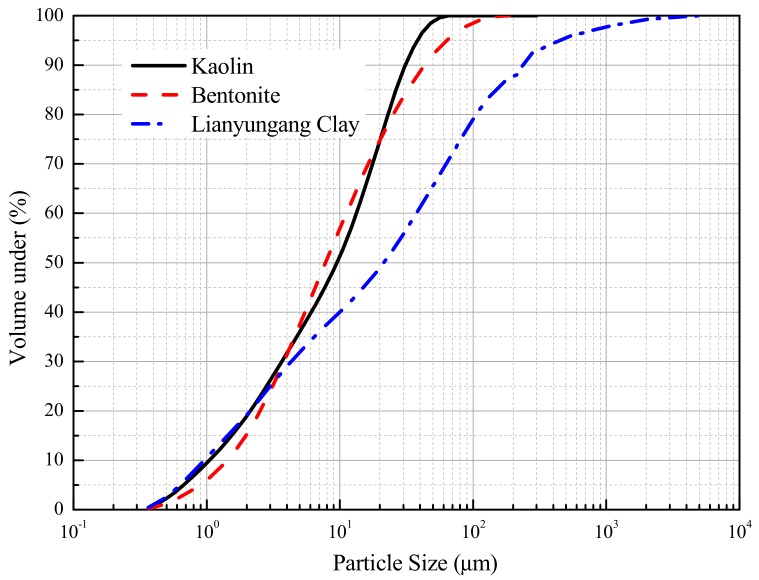
Particle size distribution curves of the considered materials.

**Figure 2 materials-12-01671-f002:**
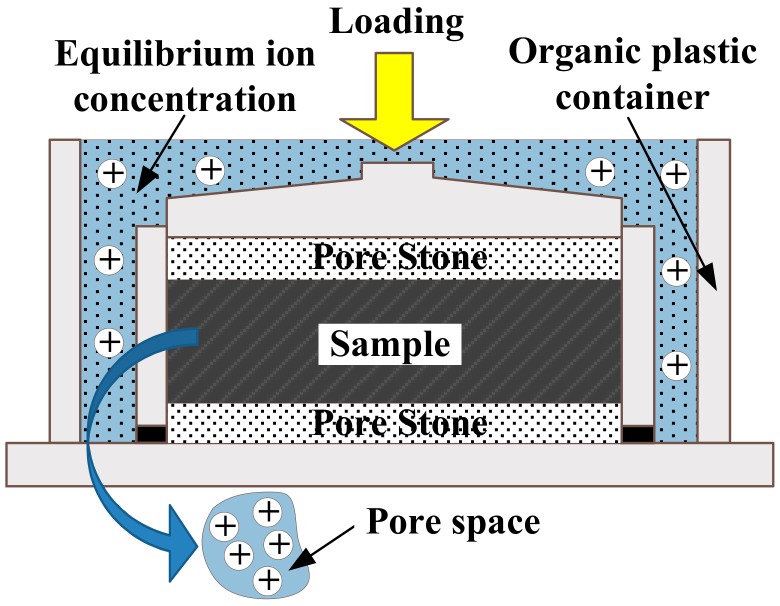
Sketch of the equipment for the oedometer test.

**Figure 3 materials-12-01671-f003:**
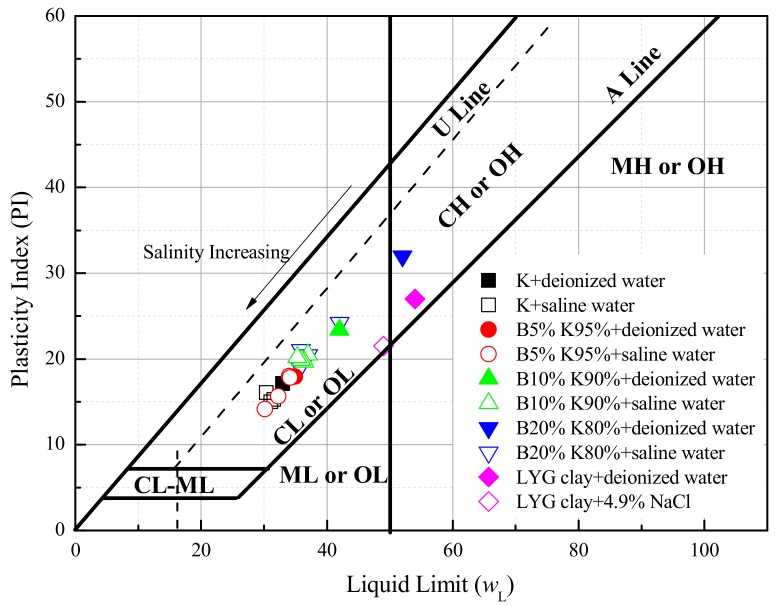
Liquid limits and plastic index comparison in plasticity chart.

**Figure 4 materials-12-01671-f004:**
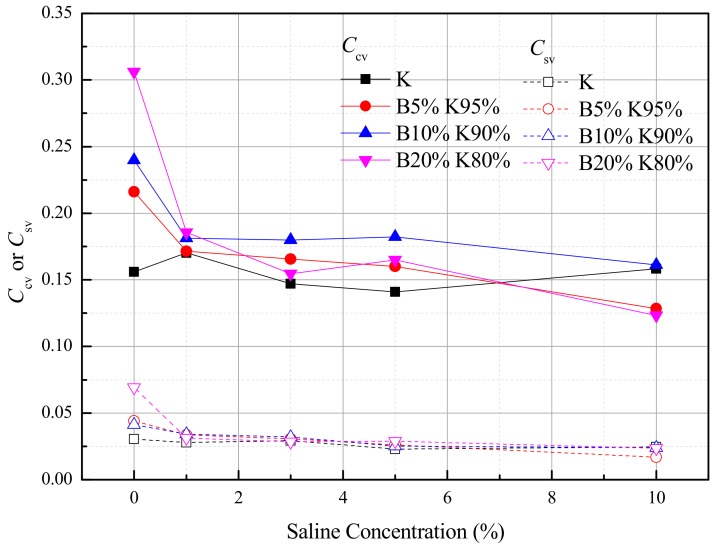
The volume compression index *C*_cv_ and the swelling index *C*_sv_ of the clays changed with sodium chloride concentrations.

**Figure 5 materials-12-01671-f005:**
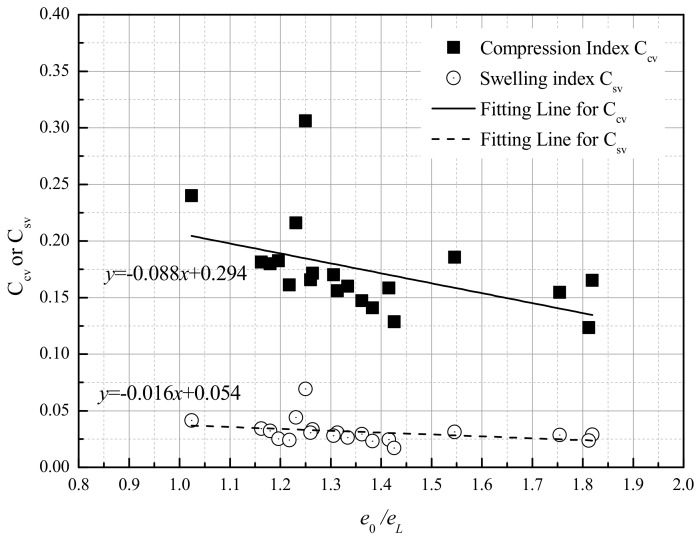
The relationship between liquid limits *e*_0_/*e*_L_ and *C*_cv_ or *C*_sv_.

**Figure 6 materials-12-01671-f006:**
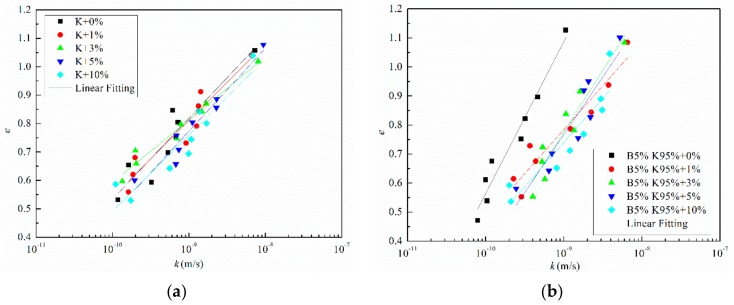
The relationships between log*k* and void ratio *e* for samples with different salinities: (**a**) K; (**b**) B5% K95%; (**c**) B10% K90%; (**d**) B20% K80%.

**Figure 7 materials-12-01671-f007:**
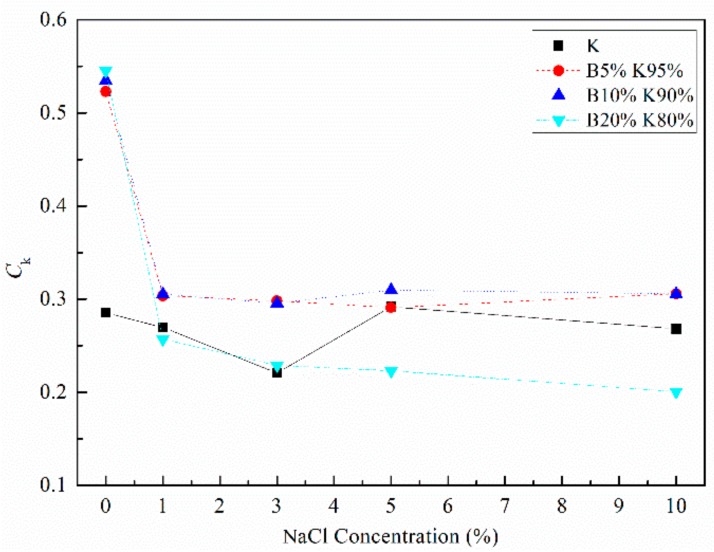
The parameter *C_k_* vs. NaCl concentration.

**Figure 8 materials-12-01671-f008:**
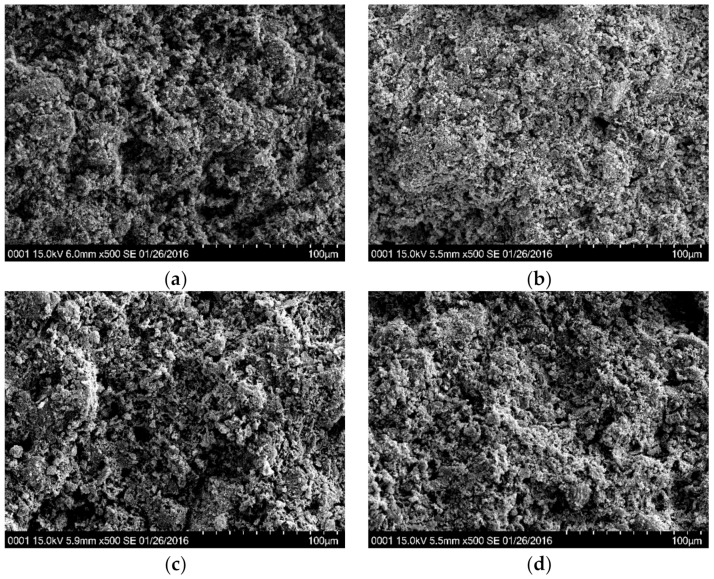
The SEM images of samples magnified 500 times. (**a**) KMicro + 0%; (**b**) KMicro + 5%; (**c**) B20%K80%Micro + 0%; (**d**) B20%K80%Micro + 5%.

**Figure 9 materials-12-01671-f009:**
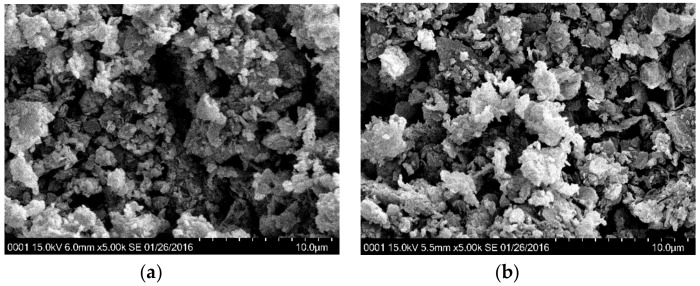
The SEM images of samples magnified 5000 times. (**a**) KMicro + 0%; (**b**) KMicro + 5%; (**c**) B20%K80%Micro + 0%; (**d**) B20%K80%Micro + 5%.

**Figure 10 materials-12-01671-f010:**
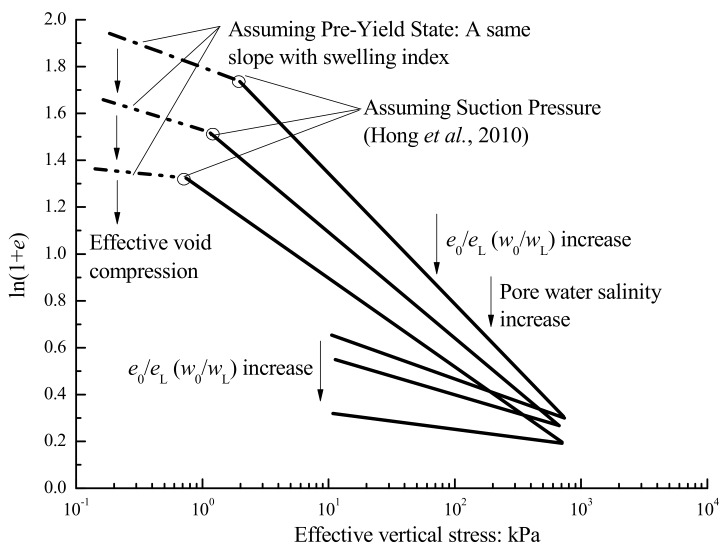
The conceptual diagram to specify the relationship between *e*_0_/*e*_L_ and *C*_cv_ or *C*_sv_.

**Table 1 materials-12-01671-t001:** Initial state of the samples.

Sample ID	Initial Water Content *w*_0_ (%)	Initial Void Ratio *e*_0_
K	43 ± 1	1.08 ± 0.01
B5%K95%	42 ± 1	1.10 ± 0.02
B10%K90%	43 ± 1	1.13 ± 0.02
B20%K80%	65 ± 1	1.78 ± 0.02
B20%K80%Micro	65 ± 1	1.78 ± 0.02
KMicro	43 ± 1	1.08 ± 0.01

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
