# Peer review of "Explanation of the Influence of Sodium Chloride Solution on Volume Deformation and Permeability of Normally Consolidated Clays"

_materials, 2019, doi:10.3390/ma12101671_

Round 1

Reviewer 1 Report

The topic of this manuscript is of great interest for many researchers, readers of your valuable Symmetry.

This paper  focuses  on the clay mineral composition and salinity effect on volume  compression-swelling behavior, permeabilities and microstructures of saturated remolded soil with  a water content more than liquid limits.

I read the manuscript carefully and patiently.  The work is well written and organized. 

However, I think the authors have to make the following corrections:

1)     The author need to emphasize more clearly the contribution of the manuscript from a scientific point of view in Abstract and in Conclusion;

2)     Arrange in page the Eqs., put point at the end and the number which counts the Eq. on the same line in the end;

3)      Use the same format and characters for all references, for example, see: [7], [24],  [30];

4)      Line 10 delete one comma, there are two: ,,

5)   The references are adequate, necessary.  I think, the author must strengthen the References section with the references that use the same technique, to make the technique used more plausible, for instance:

- A Study on the Susceptibility to SCC of 7050 Aluminum Alloy by DCB Specimens. Materials, 9, 884-894, (2016).

- Mathematical modelling of the interface crack propagation in a pre-stressed fiber

reinforced elastic composite,  Computational Materials Science 45 (3), 684-692, (2009);

- Influence of Zeolite Coating on the Corrosion Resistance of AZ91D Magnesium Alloy, Materials,  7(8), 6092-6104, (2014).

Author Response

Dear Reviewer

This paper was improved based on your comments and suggestions. The one-on-one responses were listed as follows:

The topic of this manuscript is of great interest for many researchers, readers of your valuable Symmetry. This paper  focuses  on the clay mineral composition and salinity effect on volume  compression-swelling behavior, permeabilities and microstructures of saturated remolded soil with  a water content more than liquid limits.

I read the manuscript carefully and patiently.  The work is well written and organized. 

Thank you very much for your appreciation and your useful advices.

However, I think the authors have to make the following corrections:

1)     The author need to emphasize more clearly the contribution of the manuscript from a scientific point of view in Abstract and in Conclusion;

To clarify the contribution of the manuscript from a scientific point of view, the Abstract and Conclusion were improved based on reviewer’s suggestion. The revisions were marked in manuscript.

2)     Arrange in page the Eqs., put point at the end and the number which counts the Eq. on the same line in the end;

Many thanks, the formats of Eqs. were revised based on your suggestion.

3)      Use the same format and characters for all references, for example, see: [7], [24],  [30];

The formats of all references were kept consistence, and the revisions were marked in red.

4)      Line 10 delete one comma, there are two: ,,

Done.

5)   The references are adequate, necessary.  I think, the author must strengthen the References section with the references that use the same technique, to make the technique used more plausible, for instance:

- A Study on the Susceptibility to SCC of 7050 Aluminum Alloy by DCB Specimens. Materials, 9, 884-894, (2016).

- Mathematical modelling of the interface crack propagation in a pre-stressed fiber

reinforced elastic composite,  Computational Materials Science 45 (3), 684-692, (2009);

- Influence of Zeolite Coating on the Corrosion Resistance of AZ91D Magnesium Alloy, Materials,  7(8), 6092-6104, (2014).

Many thanks, the above references that use the same technique were cited in the manuscript.

Tongwei Zhang and Shijun Wang

2019.05.14

Reviewer 2 Report

Dear Editor,

dear Authors,

An effort was made by authors to present a study on the explanation of the sodium chloride solution influences on volume  deformation and permeability of normally consolidated clays.

After reviewing the paper I could say that the present work is well organized and the obtained results could be considered suitable for the scientific community. Hence, I would recommend this research to be published in the Materials journal but after accomplishing the following suggestions:

1. Too little literature data are presented in the Introduction part. It is preferable to be enhanced with more citations regarding modern applications of clays, in order to increase the impact of the current work. Some papers like the following is suggested:

Polymers 2019, 11(4), 730; https://doi.org/10.3390/polym11040730.

2. Please correct the phrases stated in lines 28 and 37-38.

3.  Provide more details in the Experimental Method part regarding the Laser-based techniques (apparatus model, experimental conditions) used for particle-size measurement.

Author Response

Dear Reviewer

This paper was improved based on your comments and suggestions. The one-on-one responses were listed as follows:

An effort was made by authors to present a study on the explanation of the sodium chloride solution influences on volume  deformation and permeability of normally consolidated clays.

After reviewing the paper I could say that the present work is well organized and the obtained results could be considered suitable for the scientific community. Hence, I would recommend this research to be published in the Materials journal but after accomplishing the following suggestions:

Many thanks for your appreciation.

1. Too little literature data are presented in the Introduction part. It is preferable to be enhanced with more citations regarding modern applications of clays, in order to increase the impact of the current work. Some papers like the following is suggested:

Polymers 201911(4), 730; https://doi.org/10.3390/polym11040730.

Thanks for your suggestion, the modern applications of clays such as the above paper was cited in the Introduction part.

Recently, the organ modified montmorillonite was used as fillers of nanocomposite resins [1].

2. Please correct the phrases stated in lines 28 and 37-38.

The phrases stated in lines 28 and 37-38 were corrected.

Finally, the change of volume deformations and permeabilities induced by sodium chloride solution was discussed based on ‘suction pressure’ and initial void compression in micro-level.

Marine soft clays are quaternary sediments and widely distributed in southeastern China.

3.  Provide more details in the Experimental Method part regarding the Laser-based techniques (apparatus model, experimental conditions) used for particle-size measurement.

The details in the Laser-based techniques used for particle-size measurement were added in the manuscript.

The procedure is summarized as follows: 1) thirty grams of air-dried soil was stirred into 200 mL distilled water or sodium chloride solution (a mass concentration of 5%) and then soaked for 16 hours; 2) the suspension was then slowly added dropwise into the container of the laser analyser (MAF5000, Malvern Instruments Ltd., U.K.), which was filled with 800 mL distilled water; 3) this test was terminated when the required attenuation (10~20%) of the laser beam was reached; 4) based on the relationship between the scattering angle, particle radius and angular distribution of light intensity, the particle size distributions of the fine particles were obtained. For the limitation of light visibility, particles smaller than 0.3 μm could not be detected.

Tongwei Zhang and Shijun Wang

2019.05.14

Reviewer 3 Report

Good job!

Author Response

Dear Reviewer

Many thanks for your appreciation.

Tongwei Zhang and Shijun Wang

2019.05.14
